# Plasmid-DNA Delivery by Covalently Functionalized PEI-SPIONs as a Potential ‘Magnetofection’ Agent

**DOI:** 10.3390/molecules27217416

**Published:** 2022-11-01

**Authors:** René Stein, Felix Pfister, Bernhard Friedrich, Pascal-Raphael Blersch, Harald Unterweger, Anton Arkhypov, Andriy Mokhir, Mikhail Kolot, Christoph Alexiou, Rainer Tietze

**Affiliations:** 1Department of Otorhinolaryngology-Head and Neck Surgery, Section of Experimental Oncology and Nanomedicine (SEON), Else Kroener-Fresenius-Stiftung-Professorship, Universitätsklinikum Erlangen, 91054 Erlangen, Germany; 2Department of Chemistry and Pharmacy, Organic Chemistry II, Friedrich-Alexander-University Erlangen-Nürnberg, 91058 Erlangen, Germany; 3Department of Biochemistry and Molecular Biology, School of Neurobiology, Biochemistry &Biophysics, Tel-Aviv University, Tel-Aviv 6997801, Israel

**Keywords:** superparamagnetic iron oxide nanoparticles (SPIONs), surface functionalization, ligand exchange, plasmid-DNA, magnetofection, transfection, cytotoxicity, pentafluorophenyl ester

## Abstract

Nanoformulations for delivering nucleotides into cells as vaccinations as well as treatment of various diseases have recently gained great attention. Applying such formulations for a local treatment strategy, e.g., for cancer therapy, is still a challenge, for which improved delivery concepts are needed. Hence, this work focuses on the synthesis of superparamagnetic iron oxide nanoparticles (SPIONs) for a prospective “magnetofection” application. By functionalizing SPIONs with an active catechol ester (CafPFP), polyethyleneimine (PEI) was covalently bound to their surface while preserving the desired nanosized particle properties with a hydrodynamic size of 86 nm. When complexed with plasmid-DNA (pDNA) up to a weight ratio of 2.5% pDNA/Fe, no significant changes in particle properties were observed, while 95% of the added pDNA was strongly bound to the SPION surface. The transfection in A375-M cells for 48 h with low amounts (10 ng) of pDNA, which carried a green fluorescent protein (GFP) sequence, resulted in a transfection efficiency of 3.5%. This value was found to be almost 3× higher compared to Lipofectamine (1.2%) for such low pDNA amounts. The pDNA-SPION system did not show cytotoxic effects on cells for the tested particle concentrations and incubation times. Through the possibility of additional covalent functionalization of the SPION surface as well as the PEI layer, Caf-PEI-SPIONs might be a promising candidate as a magnetofection agent in future.

## 1. Introduction

Magnetic drug delivery vehicles with a non-specific enrichment behavior have been successfully investigated for potential medical applications in recent years to overcome the limitations of systemic administration with all its inherent disadvantages, such as low bioavailability and severe side effects [1,2]. Since the tumor vascularization is leaky and characterized by abnormal branching, disrupted basement membrane and enlarged interendothelial gaps, the extravasation of particulate materials from the vessel into the tumor is facilitated (enhanced permeability and retention effect, EPR effect) [3,4]. Thus, nanoparticles preferentially accumulate in tumors, resulting in higher tumoral drug accumulation and lower healthy tissue concentrations compared to the distribution of their pure small-molecule counterparts [5].

In addition to classical active agents based on small molecules, new concepts of action are playing an increasingly important role. In this regard, gene therapy is an exciting area of therapeutic research. A major benefit of this concept is that toxicity of healthy tissue may be avoided if suitable targeted delivery and tumor-specific expression strategies are employed. Due to their closed ring structure and the resulting stability, plasmid-DNA (pDNA) in particular are a crucial factor as vehicles for the transport of therapeutic genes [6,7]. Using this strategy, tumor cells are transfected with such vectors to reprogram the metabolism of the thus ‘hijacked’ tumor cell with the purpose of its own destruction. To bring pDNA into eukaryotic cells, transient opening of pores is necessary to allow uptake of the material, a process referred to as transfection. Transfection is typically performed by electroporation or transfection reagents, which fuse with the cell membrane and deposit their cargo inside. However, “magnetofection”, a transfection method that uses magnetic forces to bring pDNA into the target cells, is also available. pDNA is loaded onto magnetic nanoparticles and the magnetic force pulls the particles together with the pDNA into the target cells, where the cargo is released.

Currently, most of the work on this topic uses encapsulation in the form of micelles or particles with cationic polymer surfaces, to which the opposite charged pDNA can bind. Objects commonly developed for this purpose are micelles containing both pDNA and superparamagnetic iron oxide nanoparticles (SPIONs) or SPIONs with a cationic polymer layer adsorbed onto the particle surface [8,9]. In medical research, SPION systems were intensively investigated in the last decade for applications as contrast agents for magnetic resonance imaging (MRI) [10,11,12], as well as magnetic drug targeting systems [13,14,15]. In general, it could be demonstrated that SPIONs (with various coatings and functionalization) are spontaneously taken up by cells. Moreover, engulfment was discriminated from attachment to the cell surface by co-ingestion of lucifer yellow, a plasma-membrane impermeable dye [16]. Spontaneous uptake of iron oxide nanoparticles was also used to bring covalently bound DNA into cells [17]. Iron oxide nanoparticles with loading of a luciferase reporter construct transfected cells with similar efficacy as commercially available transfection reagent Lipofectamine (Lipo) [18]. Electroporation or applications of Lipo would not work for clinical applications. The balance between effective transfection and toxicity of the pDNA vehicles must be carefully balanced.

In our work, we have attempted to address this trade-off by developing a SPIONs system that is both capable of delivering pDNA into target cells, while being biocompatible. Furthermore, the SPION system should give the opportunity to be chemically modified in the future for e.g., improved cellular uptake or biocompatibility. Therefore, it consists of a polyethylenimine (PEI) layer that is covalently linked to a primary catechol layer. The pentafluorophenyl activated ester of caffeic acid (CafPFP) was deposited on the SPION surface by a ligand exchange reaction. To our knowledge, SPIONs which are surface functionalized with the combination of caffeic acid as an anchor and covalently bound PEI for pDNA complexation represent a novel system for a potential gene delivery application. The pDNA, which carries a GFP sequence as a model, is strongly adsorbed via electrostatic interaction with the PEI polymer.

## 2. Results

### 2.1. Synthesis of Covalently PEI-Functionalized SPIONs

In the first step to covalently bind PEI to the SPION surface, a ligand exchange was performed. A successful ligand exchange, in which water-soluble citrate ions were replaced with water-insoluble CafPFP on the SPION surface, was indicated as CafPFP-SPIONs were dispersible in ethanol (EtOH), but no longer in pure H_2_O. Further, CafPFP-SPIONs were dispersible in H_2_O, if they had been prewetted with low quantities of EtOH (~5% (*v*/*v*) EtOH of total H_2_O).

The ligand exchange did not show a significant influence on the Z-average (Z-avg.) or the polydispersity index (PDI) of the SPION system. The Z-avg. changed from 52 ± 2 nm for citrate-stabilized SPIONs (Cit-SPIONs) to 54 ± 4 nm for CafPFP-SPIONs (Figure 1a) with corresponding PDI values of 0.207 ± 0.026 and 0.224 ± 0.035 (Figure 1b). Similarly, no significant changes were observed in the ζ-potential at pH 7.3 of both SPION systems (Figure 1c) with values of −38.9 ± 2.5 mV (Cit-SPIONs) and −38.6 ± 4.9 mV (CafPFP-SPIONs). Analyzing the volumetric susceptibility for dispersions of 1 mg Fe/mL (Figure 1d), showed a slight decrease in the magnetic properties of CafPFP-SPIONs (4.82 × 10^−^^3^ ± 0.08 × 10^−^^3^) after the ligand exchange compared to Cit-SPIONs (5.15 × 10^−3^ ± 0.25 × 10^−3^). Thus, particle parameters were only slightly influenced by the ligand exchange. This was intended, as the main goal was to change the chemical properties of the particle surface. Investigating the surface chemistry of both systems using FTIR revealed significantly different spectra (Figure 1e). Normalized on the peak correlated to the F-O bond vibrations at wavenumbers of ~560 cm^−1^ [19,20], peaks representing the COO^-^ vibrations of citrate at 849, 906, 1384 and 1582 cm^−1^ [21,22,23] drastically decreased their intensities on CafPFP-SPIONs in comparison to Cit-SPIONs. After the ligand exchange, characteristic peaks appeared that can be related to vibrations in the PFP-ring system at 1008, 1104 and 1522 cm^−1^ [24,25,26] as well as a broader band at ~1729 cm^−1^ which can represent the active ester group [24,25,26]. Comparing spectra of CafPFP-SPIONs to pure CafPFP further indicates (Figure A1a) that the ligand exchange was successful in functionalizing the SPION surface with the active ester CafPFP.

Covalently binding 25 kDa PEI to the activated ester of CafPFP-SPIONs significantly increased the Z-avg. to 86 ± 2 nm (Figure 1a) of the now modified Caf-PEI-SPIONs while the corresponding PDI value stayed comparable at a value of 0.211 ± 0.016 (Figure 1b). The inversion of the ζ-potential at pH 7.3 to a value of 39.3 ± 2 mV indicates the binding of PEI onto the CafPFP surface of the SPIONs (Figure 1c). This positive charge around the SPIONs is essential for strongly complexing pDNA to the surface. Regarding the volumetric susceptibility, an additional decrease to a value of 4.43 × 10^−3^ ± 0.14 × 10^−3^ was observed (Figure 1d). FTIR spectra furthermore suggest a successful covalent binding to the activated ester (Figure 1e). Peaks previously correlated to the activated ester (~1729 cm^−1^) as well as to the PFP-ring (1008, 1104 and 1522 cm^−1^) disappeared completely in the spectra of Caf-PEI-SPIONs. Moreover, the appearance of a shoulder at ~1640 cm^−1^ (C=O, Amide I [21]) indicates the covalent modification of the surface with PEI through an amide bond release of the PFP-ring. Compared to the spectrum of pure PEI, an increase in intensity of the above-mentioned shoulder can be seen (Figure A1b), which additionally suggests the chemical modification of PEI to the SPION surface through amide bonds.

### 2.2. Binding of pDNA onto Caf-PEI-SPION System

When complexing Caf-PEI-SPIONs with varied amounts of pDNA to achieve pDNA/Fe ratios of 0.0–7.5 wt%, stable Z-avg. values around 78 nm for produced pDNA-SPIONs were observed until 2.5 wt% pDNA (Figure 2a). Similar to the Z-avg., PDI values increased with pDNA amounts higher than 2.5 wt% from around 0.200 to 0.207 and 0.226 for 5.0 and 7.5 wt% pDNA, respectively, as depicted in Figure 2b as well as in Table 1. By investigating the ζ-potential at pH 7.3 (Figure 2c), a likewise deviation at higher pDNA amounts compared to lower amounts were noticed. The ζ-potential stayed stable with values between 40.6 to 41.5 mV up to 2.5 wt% pDNA, before the values slightly decreased for 5.0 and 7.5 wt% pDNA to 39.0 ± 0.9 and 37.8 ± 1.1 mV, respectively. The SEM image in Figure 2d exemplarily shows an aggregate of pDNA-SPIONs loaded with 2.5 wt% pDNA. The aggregate appears to be assembled of individual smaller particles due to its rough texture.

The investigated pDNA/Fe ratios exhibited identically high pDNA complexation to the Caf-PEI-SPION system between values from 85% up to 99%, as shown in Figure 2e. Likewise, an increase in peak intensities which can be related to pDNA (Figure A1c) at wavenumbers 963, 1063 and 1086 cm^−1^ can be observed in FTIR analysis (Figure 2g). These peaks represent the deoxyribose stretching as well as phosphate groups in pDNA [27]. The more pDNA is complexed on the particles, the higher the peak intensity. Further investigation of the binding stability of pDNA on pDNA-SPIONs using gel electrophoresis verified a strong interaction between the particles and pDNA. Free pDNA could not be detected for any weight-ratio of pDNA-SPIONs, as visualized in Figure 2f for one experiment. Triplicate experiments and control without (*w*/*o*) SPIONs are visualized in Figure A2. Pure pDNA (as used for binding on pDNA-SPIONs) showed two bands at ~3000 as well as between 6000–8000 bp. After enzymatic linearization, the pDNA is split into two parts with ~1772 and ~2954 bp, which is expected for the linearization with the BspHI restriction enzyme.

### 2.3. pDNA Transfection and Cell Viability Analysis

After investigating the particle parameters of pDNA-SPIONs when complexed with increasing amounts of pDNA, 2.5 wt% pDNA were chosen for further experiments as no significant changes in particle properties was observed up until this value. When transfecting A375-M cells with 10 ng pDNA using either pDNA + Lipofectamin (Lipo), pDNA-SPIONs or a combination of pDNA-SPIONs + Lipo, and analyzing the number of cells that in consequence produced green fluorescent protein (GFP), it was observed that pDNA-SPIONs were able to transfect 2.3% ± 0.2% and 3.5% ± 0.3% of the cells after 24 and 48 h of incubation, respectively (Figure 3a). pDNA + Lipo transfected 1.3% ± 0.7% and 1.2% ± 0.5% of the cells after 24 and 48 h of incubation, while pDNA-SPIONs + Lipo exhibited transfections of 3.6% ± 1.0% and 4.2% ± 0.8% for 24 and 48 h of incubation. pDNA-SPION reached a statistically significant higher transfection of cells after 48 h in comparison to pDNA + Lipo whereas the combination of pDNA-SPIONs + Lipo reached significantly higher values even after 24 h of incubation when compared with pDNA + Lipo. Similar findings were observed by analyzing the cells under the microscope as depicted in Figure 3c.

When treating the cell with only 5 ng of pDNA (Figure A3a), the differences in samples transfected by pDNA + Lipo, pDNA-SPIONs and pDNA-SPIONs + Lipo decreased. The number of transfected cells were 0.8% ± 0.2%, 1.2% ± 0.7% and 2.1% ± 0.9% at 24 h for pDNA + Lipo, pDNA-SPIONs and pDNA-SPIONs + Lipo, respectively, and 0.7% ± 0.3%, 1.6% ± 0.7% and 1.9% ± 0.7% for 48 h.

Concerning the control sample, in which only medium, pure pDNA without transfecting agent as well as pDNA + PEI as transfecting agent were given to the cells, only 10 ng pDNA + PEI exhibited the slightest increase in fluorescence signal (0.2% ± 0.1%) after 48 h of incubation, as depicted in Figure A4.

Analyzing the viability of cells treated with 10 ng pDNA after 24 h and 48 h (Figure 3b) showed no statistically significant impact of pDNA + Lipo, pDNA-SPIONs as well as pDNA-SPIONs + Lipo for both investigated timepoints. At 24 h, the amount of viable cells (AxV-PI-) stayed comparable to the negative control (Medium, 82.3% ± 11.3%) with values of 82.0% ± 11.0%, 83.2% ± 6.9% and 80.1% ± 6.6% for pDNA + Lipo, pDNA-SPIONs and pDNA-SPIONs + Lipo, respectively. After 48 h, a general increase in cell viability (92.5% ± 1.9%) was observed, while no statistically significant decrease was found for cells treated with pDNA + Lipo, pDNA-SPIONs and pDNA-SPIONs + Lipo (91.1% ± 2.0%, 89.9% ± 3.0% and 88.2% ± 6.6%).

## 3. Discussion

This work aimed to create a superparamagnetic carrier system for transfecting cells with a loaded pDNA. Citrate-stabilized SPIONs were chosen as a base system for a two-step functionalization in order to create a SPION system with covalently bound PEI to its surface through a caffeic acid linker.

### 3.1. Functionalization of SPIONs with Covalently Bound PEI

Functionalizing Cit-SPIONs with CafPFP yielded SPIONs with comparable hydrodynamic size, PDI and ζ-potential. Their volumetric magnetic susceptibility decreased slightly after CafPFP functionalization most likely due to a certain degree of surface oxidation during the heat-induced ligand exchange. As the exchange is performed in a partly aqueous solution, using an inert atmosphere most likely reduces the amount of surface oxidation but cannot prevent it completely.

A successful ligand exchange was observed by CafPFP-SPIONs only being dispersible in organic solvents due to their fluorinated surface chemistry, which was also indicted by FTIR measurements. Prewetting CafPFP-SPIONs with EtOH led to their dispersibility in H_2_O, presumably due to EtOH acting as an amphiphile surrounding the SPIONs.

Activating the carboxyl group of the caffeic acid molecule by functionalizing it with the PFP-ring, showed two advantages for the SPION system. Firstly, the ligand exchange without the activated ester of the caffeic acid could not be achieved. Functionalization with pure caffeic acid led to large SPION agglomerates that were barely dispersible in H_2_O or organic solvents. Amstad et al. [28] found an enhanced surface binding onto iron oxides if an electron withdrawing group (e.g., a nitro group) was added to the catechol system. Binding the highly electron withdrawing PFP-ring to caffeic acid might have a similar effect. Hence, the binding affinity of CafPFP toward the SPION surface is higher compared to pure caffeic acid, which enables the successful ligand exchange and displacement of citrate ions by CafPFP. As a second advantage, the CafPFP functionalization causes a highly reactive surface for strongly binding further amine-containing molecules. Lawrence and Emrick [24] demonstrated that PFP-ester functionalized gold nanoparticles could be covalently bound to various amine-terminated nanoparticle systems such as SPIONs.

Conjugating PEI to the CafPFP-SPION system increased the hydrodynamic size of the system slightly to 86 nm, which might be caused by some small degree of chemically crosslinking two particles by one molecule of PEI. However, the particle size is still in the suitable range for their cellular internalization [29,30]. Additionally, the thereby created positive surface charge (+39 mV) is known to increase particle uptake into cells [31]. This is an important factor for pDNA delivery as the pDNA has to be released inside the cell for its transportation towards and transcription in the nucleus. Covalently bound PEI-SPIONs on the basis of citrate-stabilized SPIONs synthesized by Zhou et al. exhibited higher hydrodynamic sizes of 165 nm and a slightly lower surface charge of +22.5 mV, while still being strongly taken up by B16-F10 cells to concentrations of 15 pg Fe/cell without an external magnetic field [32].

### 3.2. pDNA Binding onto the SPION System

After complexation with various weight ratios of pDNA/Fe, Caf-PEI-SPION parameters were found to be unchanged until 2.5 wt% pDNA, while binding almost all added pDNA to their surface. Adding larger amounts of pDNA (up to 7.5 wt%) to the SPIONs still resulted in high binding efficiencies. Xiao et al. even reported pDNA binding of > 90% for pDNA/SPION weight ratios of 8/1 [33]. Once complexed with pDNA, however, hydrodynamic sizes of the complexes synthesized by Xiao et al. increased to 300 nm and larger with increasing pDNA/SPION ratios from 0.5/1 to 16/1 [33]. Likewise the covalently modified PEI-SPIONs reported by Zhou et al. faced challenges in keeping the hydrodynamic size of the system < 200 nm after pDNA binding [32]. At 7.5 wt% pDNA on Caf-PEI-SPIONs, the hydrodynamic sizes were still <200 nm, even though the reproducibility of the system decreased drastically. At 10 wt% pDNA the system collapsed and formed large aggregates due to an almost neutral ζ-potential of the pDNA-SPIONs. Even though the total amount of pDNA bound to the SPIONs was lower compared to the stated research, 2.5 wt% pDNA were chosen for cell viability and transfection experiments due to the higher reproducibly of the system and smaller particle size.

### 3.3. pDNA Transfection into A375-M Cells and Cell Toxicity

Transfecting A375-M cells with 5 ng pDNA either complexed with Lipo, Caf-PEI-SPIONs or Caf-PEI-SPIONs + Lipo indicated no statistically significant difference for any system even after 48 h of incubation. This might be explained by the amount of pDNA given to the cells being too low for a reproduceable transfection with either of the systems. Doubling the amount of pDNA to 10 ng on the other hand led to an almost 3-fold higher transfection efficiency for pDNA-SPIONs compared to Lipo after 48 h of incubation. Even the combination of SPIONs + Lipo only increased the transfection efficiency to 3.5-fold of the value of only Lipo after 48 h, which indicates the potential of the pure Caf-PEI-SPIONs as transfection system. The increase in transfection efficiency from 24 h to 48 h might be caused by the slower cellular internalization and pDNA release of only SPIONs in comparison to SPIONs + Lipo. The addition of Lipo to the system might facilitate the cellular uptake of the particles.

Even though the total transfection efficiency of the system (3.5%) needs to be improved, the percentage of seeded cells (30 k) transfected with only 10 ng of pDNA is already promising. Zhou et al. showed a transfection efficiency of 15–20% in A549 and B16-F10 cells without, and up to 37.5% with magnetic attraction of the SPIONs, while using 9 times more pDNA per seeded cell for transfection (300 ng pDNA on 100 k seeded cells) [32]. Similarly, Kamau et al. reported transfection efficiencies in 293T cells of 14% without and up to 73% with magnetic attraction when incubating cells with 5 µg of pDNA bound to SPIONs [34]. These results display the potential that magnetofection can have. For Caf-PEI-SPIONs, this magnetofection potential has to be explored in future research whilst simultaneously increasing the binding of pDNA to the system without enlarging the particle size drastically. Furthermore, emphasis has to be placed on the biocompatibility of Caf-PEI-SPIONs. Cytotoxic effects of positively charged particles have to be faced at a certain concentration [35]. As shown in Figure 3b, no cytotoxic effects on A375-M cells were detectable after 48 h of incubation with an iron concentration of 1.3 µg/mL. Increasing the iron concentration and thus the pDNA amount during transfection might further enhance the transfection efficiency of the system.

For both mentioned challenges, higher pDNA transfection rates and enhanced biocompatibility, Caf-PEI-SPIONs are an encouraging system for future research. The active esters on the surface of CafPFP-SPIONs might e.g., allow for a co-functionalization with a second amine-containing molecule during the binding of PEI to the SPIONs. This could be used in the future to enhance biocompatibility and pDNA binding. Additionally, a chemical post-functionalization of the already bound PEI molecules on the particle surface might be an option to improve the system as well without drastically changing the particle parameters. This post-functionalization could be achieved with facile and already existing protocols such as EDC/NHS (1-ethyl-3-(3-dimethylaminopropyl)carbodiimide/N-hydroxysuccinimide) chemistry. Hence, Caf-PEI-SPIONs are a fascinating topic for exploring more modification strategies in the future to create a highly efficient magnetofection system.

## 4. Materials and Methods

### 4.1. Materials

Iron (III) chloride hexa-hydrate (99%), iron (II) chloride tetra-hydrate (99%) and triethylamine (99%) were purchased from Merck (Darmstadt, Germany). Acetic acid (100%), acetone (99.9%), sodium acetate (99%), agarose GTQ, ammonia (NH_3_, 25%), bromophenol blue, caffeic acid (Caf, 98%), sodium citrate (Cit, 99%), dichloromethane (99.5%), N,N-Dimethylformamide (DMF, 99.8%), 1,4-dithiothreitol (DTT), ethanol (EtOH, 99.5%), ethylenediaminetetraacetic acid (EDTA, 99%), ethyl acetate (99.5%), glycerol, 1 M hydrochloric acid, kanamycin A, magnesium sulphate (99%, dry), nitric acid (65%), pentafluorophenyl trifluoroacetate (98%), petrol ether, pyridine (95%), silica gel 60, sodium hydroxide, sodium dodecyl sulfate sodium lauryl sulfate (SDS) and tris(hydroxymethyl)aminomethane (TRIS base, 99.9%) were purchased from Carl Roth (Karlsruhe, Germany). 4-(2-hydroxyethyl)-1-piperazineethanesulfonic acid (HEPES, 99.5%), LB Broth (Miller), polyethyleneimine (PEI, branched, M_w_ ~ 25,000) and propidium iodide (PI, 94%) were purchased from Sigma-Aldrich (Taufkirchen, Germany). 7-Aminoactinomycin D (7-AAD) was purchased from BioLegend (San Diego, CA, USA). RPMI 1640 medium, Hoechst 33,342 (Hoe), Annexin A5 allophycocyanin (APC) conjugate (AxV), β-Mecaptoethanol (50 mM), penicillin/streptomycin, and Lipofectamine 2000 (1 mg/mL) were purchased from Thermo Fisher (Waltham, MA, USA). Fetal calf serum (FCS) was purchased from Biochrom (Berlin, Germany). Trypsin/EDTA (0.05%/0.02% in phosphate-buffered saline) was purchased from PAN-Biotech (Aidenbach, Germany). A375-M cells were purchased from ATCC (Manassas, VA, USA). Ringer’s solution was purchased from Fresenius Kabi (Bad Homburg, Germany). BspHI enzyme kit for DNA modification was purchased from New England BioLabs (Frankfurt am Main, Germany). DNA ladder peqGOLD (100–10,000 bp) was purchased from VWR (Radnor, PA, USA) Deionized water was produced using a Merck Milli-Q purification system. All reagents were used without further purification.

### 4.2. Synthesis of CafPFP

Caffeic acid pentafluorophenyl ester (CafPFP) was synthesized by following the procedure of Williams et al. [36]. Briefly, caffeic acid was dissolved in 10 mL DMF at a concentration of 79 mg/mL. A volume of 0.570 mL of pyridine was added to the solution while stirring. To create an activated ester, 1.21 mL of pentafluorophenyl trifluoroacetate was added. The reaction proceeded for 2 h at 23 °C with stirring. Afterwards, the reaction solution was diluted using 30 mL of dichloromethane and everything was given to a separating funnel. Using 5× 25 mL of 1 M HCl, the reaction mixture was washed by discarding the upper aqueous phase. The organic phase was dried over magnesium sulfate powder. After filtering the solution, it was concentrated using a rotary evaporator (Hei-Vap Precision, Heidolph, Germany). A chromatography column was filled with silica gel as well as a solvent ratio of petroleum ether/ethyl acetate 3/2 for chromatography purification of the crude product. Again, a rotary evaporator was used to dry the final product. The CafPFP powder was stored under light protection and left dry until further use.

### 4.3. Synthesis of Citrate-Stabilized SPIONs (Cit-SPIONs)

Citrate-stabilized SPIONs (Cit-SPIONs) were synthesized following the protocol described by Mühlberger et al. [37]. In short, iron (II) and iron (III) salts were mixed in H_2_O to create a solution with an iron concentration of 13.2 mg/mL. Under argon protection, a 25% NH_3_ solution was quickly added into the vigorously stirred iron salt solution. The precipitated dark SPION dispersion was allowed to grow for 10 min at 23 °C. Afterwards, 15 mL sodium citrate solution containing 293.3 mg/mL was injected to the SPIONs while being stirred at 400 rpm. The reaction dispersion was heated to 90 °C and was mildly refluxed for 30 min.

After cooling to room temperature, the Cit-SPIONs were magnetically removed from the reaction supernatant. The supernatant was discarded and the Cit-SPIONs were washed magnetically using 60 mL of acetone 5 times. After drying the Cit-SPIONs from acetone, they were redispersed in H_2_O and stored at 4 °C until further use. All particle batches were synthesized in triplicate (*n* = 3).

### 4.4. Ligand Exchange to Generate CafPFP-Functionalized SPIONs (CafPFP-SPIONs)

CafPFP functionalization of Cit-SPIONs was achieved by dissolving CafPFP in pure EtOH to gain 14 mL of a 60 mM solution. The solution was mixed in an Argon atmosphere with 39 mL of 0.1 M acetate buffer pH 4, and heated to 90 °C for 10 min while stirring at 150 rpm using an overhead stirrer to dissolve any precipitated CafPFP that had formed. Next, 3 mL of a 9.33 mg Fe/mL Cit-SPION solution was rapidly injected into the reaction solution and the ligand exchange was allowed to proceed for 30 min under reflux at 90 °C and 150 rpm. Afterwards, the reaction solution was cooled to room temperature using an ice bath. CafPFP-SPIONs were collected using an external magnet and the opaque supernatant was discarded.

CafPFP-SPIONs were redispersed in 15 mL pure EtOH and transferred in equal parts into two centrifugal filters (100 kDa MWCO PES membrane, Satorius, Göttingen, Germany). A volume of 15 mL of H_2_O was added to each SPION dispersion to achieve a 50% EtOH solution prior to centrifuging for 30 min at 2000× *g*. The SPIONs were washed in this way three times before being washed three times with 7.5 mL pure EtOH using the same centrifugal procedure. Washed CafPFP-SPIONs were redispersed and removed from the filter using a total 12 mL of pure EtOH. CafPFP-SPIONs were stored in pure EtOH at 4 °C until further use. All particle batches were synthesized in triplicate (*n* = 3).

### 4.5. Covalent PEI Functionalization on CafPFP-SPIONs to Create Caf-PEI-SPIONs

Covalent binding of PEI to the activated ester of CafPFP-SPIONs was conducted according to the method of Williams et al. [36] with some modifications. Briefly, 26.25 mL of a 1.05 mM PEI solution in pure EtOH was heated to 60 °C in an Argon atmosphere while mixing at 200 rpm using an overhead stirrer. Next, 11.25 mL of a 1 mg Fe/mL CafPFP-SPION dispersion in pure EtOH was rapidly injected in the stirring PEI solution. 0.75 µL of TEA was immediately injected into the reaction solution. The reaction was allowed to proceed for 30 min at 60 °C and 200 rpm. After cooling the reaction mixture to room temperature, it was transferred into two centrifugal filters (300 kDa PES membrane, Satorius, Göttingen, Germany) in equal parts and centrifuged for 30 min at 750× *g* to remove the Caf-PEI-SPIONs from the reaction solution. The SPIONs were centrifugally washed three times with 15 mL pure EtOH in each filter tube and three times with 15 mL H_2_O in each filter tube. Washed Caf-PEI-SPIONs were redispersed in a total volume of 6 mL H_2_O. The dispersion was treated with 0.5 kJ/mL of ultrasonication to break up any aggregates that formed at the filter membrane. Caf-PEI-SPIONs were stored at 4 °C until further use. All particle batches were synthesized in triplicate (*n* = 3).

### 4.6. pDNA Binding on Caf-PEI-SPIONs

pDNA, which carried a GFP sequence, was extracted from kanamycin resistant bacteria (E. coli DH5α PEGFP-N1) after cultivation for 48 h at 37 °C while shaking at 250 rpm in LB Broth (Miller) supplemented with 150 mg/L kanamycin. The extraction was performed using a “PureLink HiPure Plasmid DNA Purification Kit” (Invitrogen, Product No. K2100-02) using the protocol provided with the kit.

Caf-PEI-SPIONs were loaded with different amounts of pDNA to investigate binding efficiency and particle stability. The weight ratio pDNA/Fe were 7.5%, 5.0%, 2.5%, 1.0%, 0.5% and 0.0%, respectively. To achieve the pDNA binding to the particles, pDNA was diluted to 60, 40, 20, 8, 4 and 0 µg/mL in a volume of 400 µL. The pDNA solution was incubated at 23 °C and 700 rpm using a mixer. Into each solution 400 µL of a 0.8 mg Fe/mL Caf-PEI-SPION dispersion was injected while mixing. The binding was allowed to proceed for 10 min. pDNA carrying Caf-PEI-SPIONs were used without further purification. pDNA carrying Caf-PEI-SPIONs are referred to as pDNA-SPIONs. All particle batches were synthesized in triplicate (*n* = 3).

### 4.7. Physicochemical Characterization

#### 4.7.1. Atomic Emission Spectroscopy (AES)

The iron content in Cit-SPIONs, CafPFP-SPIONs and Caf-PEI-SPIONs was analyzed by AES measurements (Agilent 4200 MP-AES, Agilent Technologies, Santa Clara, CA, USA). AES samples were prepared by dissolving the SPION systems in nitric acid. CafPFP-SPIONs were dried at 95 °C for 15 min to remove EtOH before adding nitric acid. The dissolved particles were diluted with H_2_O.

#### 4.7.2. Dynamic Light Scattering (DLS)

The hydrodynamic size of the SPIONs was examined by DLS using a Zetasizer Nano (Malvern instruments, Worcestershire, United Kingdom). Cit-SPIONs and Caf-PEI-SPIONs were prepared by diluting the dispersions with H_2_O, while CafPFP-SPIONs were diluted with pure EtOH. The iron concentration of the SPION systems were adjusted to 50 µg/mL. The hydrodynamic size of CafPFP-SPIONs was corrected using the Zetasizer Software, as SPIONs were dispersed in EtOH to correlate the values with systems which were dispersed in H_2_O.

#### 4.7.3. ζ-Potential Measurement

The ζ-potential of the SPIONs was determined using the Zetasizer Nano (Malvern instruments, Worcestershire, UK). SPION dispersions were diluted with H_2_O to an iron concentration of 50 µg/mL. The pH value was adjusted to 7.3 by the dropwise addition of 10 mM hydrochloric acid or 10 mM sodium hydroxide.

#### 4.7.4. Magnetic Susceptibility Measurement

To verify the magnetic properties of the synthesized SPIONs, the magnetic susceptibility was investigated by using a magnetic susceptibility meter (MS2G, Bartington Instruments, Witney, UK). The SPION dispersions were diluted with H_2_O (Cit-SPIONs and Caf-PEI-SPIONs) or with EtOH (CafPFP-SPIONs) to an iron concentration of 1 mg/mL for the measurement.

#### 4.7.5. Fourier Transform Infrared Spectroscopy (FTIR)

For the investigation of the surface chemistry, the samples were characterized with Fourier transform infrared spectroscopy. A total of 4 µL sample was dried in 1 µL portions on the crystal of a FTIR spectrometer (Alpha-P, Bruker, Billerica, MA, USA) using a stream of cold air. The measurement consisted of 128 scans ranging from 400–4000 wavenumbers/cm. OPUS software (Bruker, Billerica, MA, USA) was used for comparable background subtraction and baseline correction.

#### 4.7.6. Scanning Electron Microscopy (SEM)

The size and shape of the SPION systems were determined using a scanning electron microscope (Auriga, Zeiss, Oberkochen, Germany). The samples were prepared by freeze-drying (Alpha 1–2 LD plus, Martin Christ, Osterode am Harz, Germany) previously diluted SPION dispersions for 24 h on silicon sample holders. Images were taken at a 75,000× magnification with an acceleration voltage of 18 kV.

### 4.8. Determination of pDNA Binding Efficiency to Caf-PEI-SPIONs

#### 4.8.1. Fluorescence Measurements Using Fluorescence DNA Dye 7-AAD

To determine the amount of pDNA bound to the particle surface, pDNA concentrations in the supernatant of the SPION dispersions were analyzed. For that, pDNA-SPIONs were centrifuged at 18,000× *g* for 45 min to remove all particles from the supernatant and collect them in a pellet. 100 µL of each supernatant was removed and mixed with 10 µL 7-AAD under light protection for 10 min in a fluorescence suitable 96-well plate in order to detect any unbound pDNA. Fluorescence measurements were conducted with an absorption wavelength of 540 nm and an emission wavelength of 650 nm using a plate reader (SpectraMax iD3, Molecular Devices, Biberach an der Riß, Germany). Equally treated pDNA solutions in pure H_2_O were used as control samples.

#### 4.8.2. Gel Electrophoresis for Determining the pDNA/SPION Complex Stability

To further investigate the stability and binding strength of pDNA-SPIONs gel electrophoresis was used. To begin, 1% (*w*/*v*) agarose gels were produced by dissolving 210 mg agarose powder in 21 mL TAE buffer pH 8.3 (0.4 M Tris base, 0.01 M EDTA, 0.2 M acetic acid) using a microwave. Gels were casted to be 70 mm × 60 mm × 4.5 mm and contained 8 sample compartments. Gels were wetted with TAE buffer and stored at 4 °C until usage.

As a control, pDNA was linearized using the BspHI enzyme following the manufacturers’ instructions (New England BioLabs, Product No. R0517S).

A volume of 2 µL of 6xPPP loading dye were mixed with 10 µL of each sample. The samples consisted of 10 µg/mL pDNA as used in the pDNA-SPION system, 10 µg/mL of previously linearized pDNA, 400 µg Fe/mL pDNA-SPIONs with pDNA/Fe weigh ratios of 7.5%, 5.0%, 2.5%, 1.0% and 0.5%. In addition, 2 µL 6xPPP loading dye were mixed with 2 µL of a peqGOLD DNA ladder as well as 8 µL of H_2_O as a reference.Finally, 6 µL of the as prepared samples were pipetted into each sample compartment of the gel. Electrophoresis was conducted in TAE buffer for 35–40 min at a voltage of 100 V.

Afterwards, the gel was stained under light protection for 30 min using Gelred^®^ (Biotium, Fremont, CA, USA) diluted 1:10,000 in H_2_O. Images were taken after excitation at a wavelength of 280 nm and an emission wavelength of 600 nm.

### 4.9. pDNA Delivery Using pDNA-SPIONs and Their Biocompatability

To test the ability of pDNA-SPIONs to deliver the plasmid, which carried a GFP sequence, into cells, adherent A375-M cells were used. Detection of GFP in cells as well as cell viability after incubation with SPIONs was performed using flow cytometry. A375-M cells were cultured in RPMI 1640 medium supplemented with 10% FCS, 1% L-glutamine, 1% penicillin/streptomycin, 0.2% HEPES and 0.04% β-mercaptoethanol at 37 °C in a 5% CO_2_ atmosphere. Next, 3 × 10^4^ A375-M cells in a volume of 300 µL were seeded into each well of a 48-well plate and incubated for 16 h at 37 °C in a 5% CO_2_ atmosphere. Afterwards, 10 µL of sample were added to the cells in triplicates and the treated cells were further incubated for 24 and 48 h, respectively.

The samples added to the cells consisted of pure medium, pure pDNA, pDNA with Lipo, pDNA with PEI, pDNA-SPIONs (2.5 wt% pDNA/Fe) and pDNA-SPIONs (2.5 wt% pDNA/Fe) with Lipo. Two different pDNA concentrations, 1 and 0.5 µg/mL in the 10 µL samples were chosen for a total of 10, as well as 5 ng pDNA/well. Therefore, SPION concentrations in the 10 µL sample was adjusted to 40 as well as 20 µg Fe/mL. For the PEI control concentrations in the 10 µL sample of 4 as well as 2 µg/mL were chosen, which represent 10 wt% of iron used in the SPION sample as well as a PEI/pDNA wt% of 4.

After treating the cells with the samples for 24 or 48 h, respectively, the adherend cells were suspended using trypsin/EDTA (0.05%/0.02%) in PBS, centrifugally washed using Ringer’s solution (5 min, 300 rcf, 4 °C) and stained in 100 µL staining solution for analyzing their viability using flow cytometry by adapting the protocol of Mühlberger et al. [37]. The staining solution contained 1 µL/mL Hoe, 2 µL/mL AxV-APC and 66.7 ng/mL PI in Ringer’s solution. The staining solution was freshly prepared for each time point. After incubation of the cells for 20 min at 4 °C under light protection, a flow cytometer (Gallios, Beckman Coulter, Krefeld, Germany) was used to analyze their fluorescence.

### 4.10. Statistical Analysis

Statistical significance was examined using GraphPad PRISM 8.3.0 (GraphPad Software Inc., San Diego, CA, USA). Experiments were performed in independent triplicates. *p*-values ≤ 0.05 were considered as statistically significant.

## 5. Conclusions

This work reports and characterizes a PEI-SPION system for the potential application of magnetically transfecting cells with pDNA. The PEI-SPIONs are based on a novel CafPFP-SPION system providing a chemically easy to functionalize surface using moderate chemicals. The hydrodynamic size of the SPIONs only increased to 86 nm after PEI coating while the ζ-potential switched from −39 to +39 mV. By covalently binding PEI to the CafPFP surface, chemically post-functionalizing the strongly attached PEI coating of the SPIONs might be conceivable.

When Caf-PEI-SPIONs were complexed with pDNA, which carried a GFP sequence, these pDNA-SPIONs were able to maintain particle properties up to 2.5 wt% pDNA, whereas higher pDNA amounts started to show signs of particle agglomeration. Yet, up to 99% of the added pDNA were bound to the SPION surface even at 7.5 wt% pDNA.

Transfecting adherend A375-M cells with 2.5 wt% pDNA-SPIONs for 48 h resulted in a 3.5% transfection efficiency measured by the GFP fluorescence of the transfected cells. No cytotoxic effects of the particles on the cells were detected after 48 h of incubation.

Enhancement of the pDNA loading onto the SPION system, as well as transfection efficiency, needs to be addressed in future studies. This might be achieved by the chemical functionalization of the covalently bound PEI molecules on the particle surface either during or after PEI binding. Furthermore, magnetofection experiments have to be conducted in vitro to display the magnetic functionality of the system as well as potential biocompatibility issues, if the SPIONs are concentrated in a high amount in an external magnetic field.

## Figures and Tables

**Figure 1 molecules-27-07416-f001:**
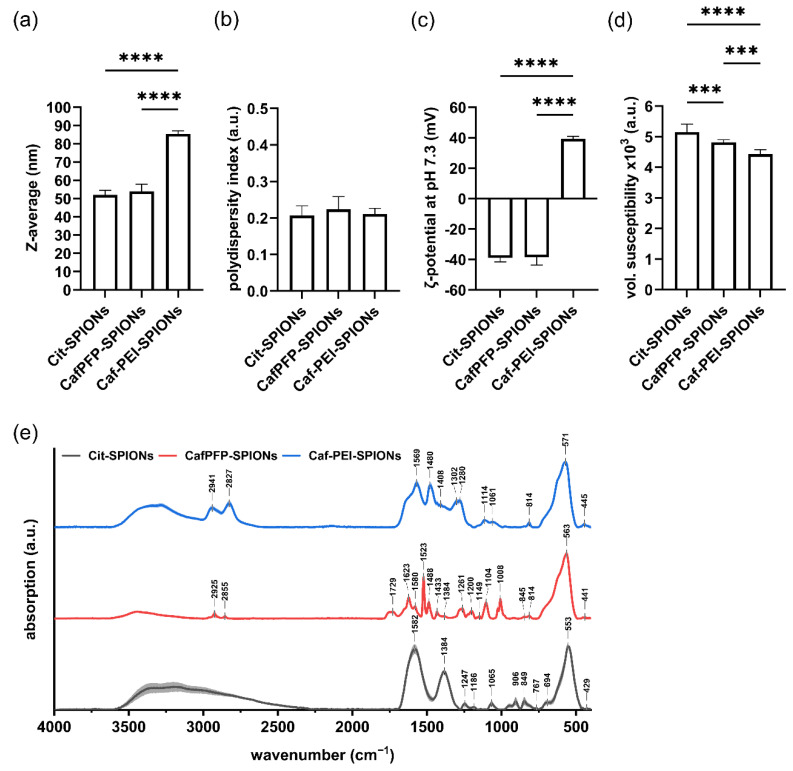
Physicochemical characterization of Cit-, CafPFP- and Caf-PEI-SPIONs. Surface modifications of SPIONs were investigated in terms of changes in their (**a**) hydrodynamic size, (**b**) polydispersity index, (**c**) ζ-potential at pH 7.3 as well as (**d**) volumetric susceptibility for 1 mg Fe/mL. Statistically significant changes were calculated with ordinary one-way ANOVA (*** for *p* < 0.001, **** for *p* < 0.0001). (**e**) FTIR spectra show the different chemical functionalities on the surfaces of Citrate-, CafPFP- and Caf-PEI-SPIONs. Spectra are normalized on the peak at ~560 cm^−1^ related to Fe-O vibrations. Shades mark the standard deviation of the particle triplicates (*n* = 3).

**Figure 2 molecules-27-07416-f002:**
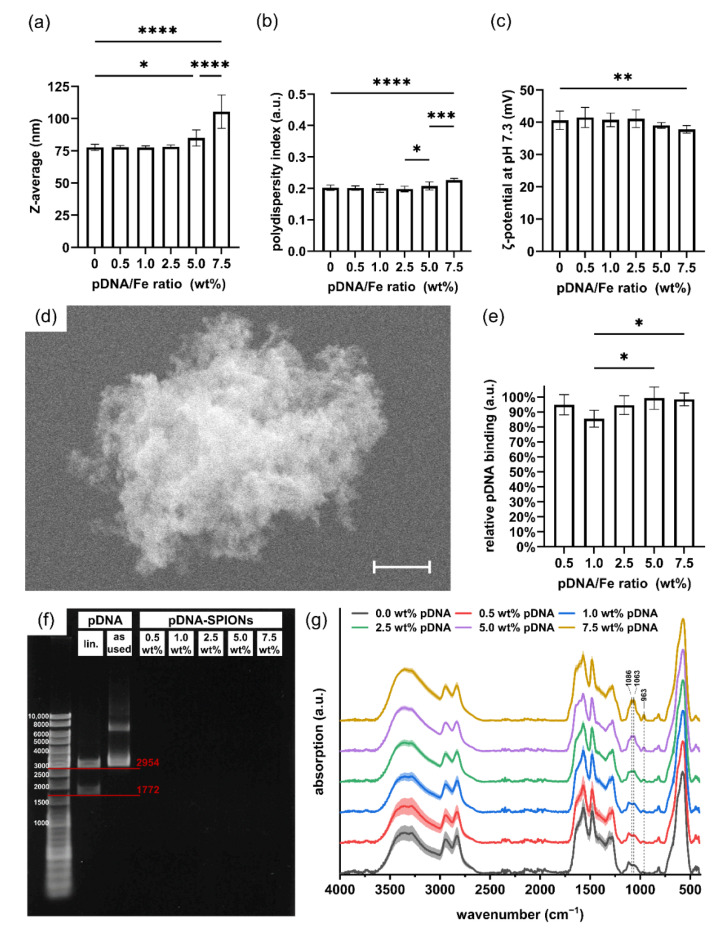
pDNA binding on Caf-PEI-SPIONs and its influence on physicochemical particle parameters. Changes in (**a**) the hydrodynamic size, (**b**) the polydispersity index and (**c**) the ζ-potential at pH 7.3 of pDNA-SPIONs with increasing weight ratio of pDNA/Fe. (**d**) SEM image of an aggregate of 2.5 wt% pDNA-SPIONs consisting of smaller individual particles. The scale bar represents 400 nm. (**e**) Binding efficiencies of various amount of pDNA mixed with Caf-PEI-SPIONs. Statistically significant changes were calculated with ordinary one-way ANOVA (* for *p* < 0.05, ** for *p* < 0.01, *** for *p* < 0.001, **** for *p* < 0.0001). (**f**) Agarose gel electrophoresis of pDNA-SPIONs showing no free pDNA for all tested pDNA/Fe weight ratios. Bands for digested, linearized (lin.) pDNA are in accordance with the expected sizes of 1772 and 2954 bp. (**g**) FTIR spectra showing the chemical changes on the particle surfaces by adding higher amounts of pDNA. Spectra are normalized on the peak at ~560 cm^−1^ related to Fe-O vibrations. Shades mark the standard deviation of the particle triplicates (*n* = 3).

**Figure 3 molecules-27-07416-f003:**
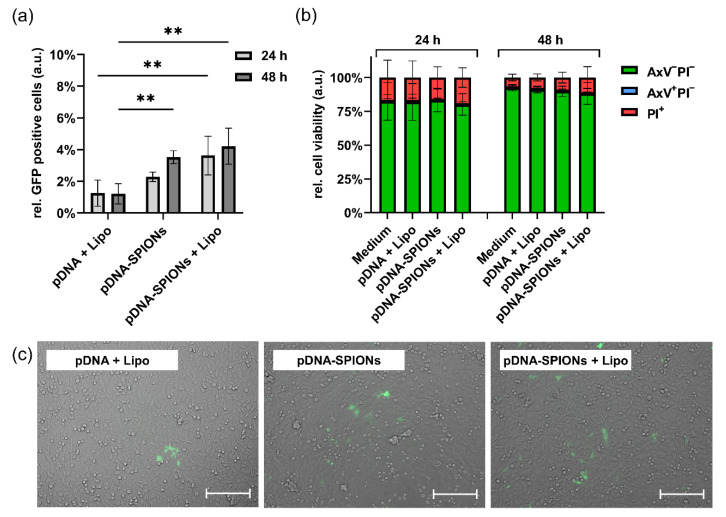
Transfection of 10 ng pDNA into A375-M cells using Lipo, pDNA-SPIONs and pDNA-SPIONs + Lipo. (**a**) Relative amount of GFP positive cells and (**b**) relative cell viability after incubation of the three different systems for 24 and 48 h. Statistically significant changes were calculated with two-way ANOVA (** for *p* < 0.01). (**c**) Exemplary microscopic images of transfected cells after 48 h of incubation. The scale bar represents 200 µm.

**Table 1 molecules-27-07416-t001:** Summary of particle parameters for pDNA-SPIONs with increasing pDNA loading.

Wt% pDNA on pDNA-SPIONs	Z-avg. in nm	PDI in a.u.	ζ-Potential at pH 7.3 in mV	Rel. pDNA Bindingin. a.u.	pDNA Load in ng/µg Fe
0.0	78 ± 2	0.202 ± 0.008	40.6 ± 2.8	-	-
0.5	78 ± 1	0.201 ± 0.006	41.5 ± 3.0	94.8% ± 4.8%	4.7 ± 0.2
1.0	78 ± 1	0.200 ± 0.012	40.7 ± 2.1	85.5% ± 4.6%	8.6 ± 0.4
2.5	78 ± 1	0.198 ± 0.008	41.1 ± 2.6	94.5% ± 5.1%	23.6 ± 1.2
5.0	86 ± 6	0.207 ± 0.012	39.0 ± 0.9	99.2% ± 6.1%	49.6 ± 3.0
7.5	105 ± 12	0.226 ± 0.006	37.8 ± 1.1	98.4% ± 3.5%	73.8 ± 2.6

pDNA: plasmid DNA; SPIONs: superparamagnetic iron oxide nanoparticles; Z-avg.: Z-average; PDI: polydispersity index; Fe: iron.

## Data Availability

Not applicable.

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
