# Peer review of "Plasmid-DNA Delivery by Covalently Functionalized PEI-SPIONs as a Potential ‘Magnetofection’ Agent"

_molecules, 2022, doi:10.3390/molecules27217416_

Round 1

Reviewer 1 Report

The paper by Stein et al. “Plasmid-DNA delivery by covalently functionalized PEI-SPIONs as a potential ‘magnetofection’ agent” deals with the transfection of cells by means of functionalized and coated SPIONs that carry a DNA load on their surface.  It is very clear and readable. Moreover, the topic is highly interesting for biomedical applications. I have a few minor comments on the content:

-       I had hoped to see some more detailed information on the final DNA load on the particles: how were the values calibrated? how accurate is the pDNA concentration determination and what are the difficulties of the methodology? You might think about validating your pDNA quantification from the supernatant, to confirm the values.

-        Line 183, “as no significant changes in particle properties was observed…”, should it be a “were” there?

-        Some data are listed as a long series of numbers. Perhaps it would be more elegant to put them in form of tables and even move some of the data to the Supporting Information.

-        In line 82 you write about coulomb interactions, but this is something you assume, but you cannot prove. You should express it in a different manner or use a reference to support it.

In general, it would enhance the strength of the authors’ message to know what is really the innovation in their system. Is it the usage of Caf, and/or the usage of PEI: what exactly have the authors done in this work that differs from previous work? Furthermore, it would also improve the paper to present and discuss the current challenges in magnetofection (is it “only” the low efficiency?), the advantages and disadvantages of the different approaches which are being employed by the scientific community and the reasons why the authors use this particular approach. Finally, I would like to have seen some discussion on what happens with all the DNA that has not interacted with the cells: why are so many particles not interacting with the cells? Do the authors have any ideas to enhance the tendency of the particles to adhere to the cell surface and allow for cell transfection rather than “only” pointing out the DNA load? It is not clear that a higher DNA load would even enhance the final magnetofection ratio.  What else could be taken into account?

Even if the last paragraph of the discussion, l. 311 to 321, is very clear and informative, it does not seem to confront the fact that such a low transfection ratio means that there is a lot of modified DNA on sophisticated SPIONS swimming around which are able to attach to everything else in the medium. If the number of transfected cells is so low, it means that most of the cells are not interested in the particles at all or vice versa. I would suggest at least to comment on that point and to express what you expect to happen with those particles: (1) do they adhere to cell surfaces and not manage to move through the cell wall? (2) do they just swim around, and eventually agglomerate and grow to become a larger body? (3) what would happen in a living system, would they mainly interact with other objects in the living system? I would be very interested if you had ideas about the impact of the non-magnetofecting particles on their surroundings. These are important open questions, which could (and perhaps should) be thought about, but go beyond the scope of this paper.

Altogether, the paper is good and should be accepted with only minor revisions which are not essential for the publication of the paper, but could improve the quality.

Author Response

Please find the file attached

Reviewer 2 Report

The authors present a new particle design for magnetofective transport agents into cells using PEI and pDNA, providing a detailed account of particle characterization before and after surface modification as well as a complete study on using the particles for magnetofection in vitro.

Authors clearly describe cell studies and quantify magnetofective efficiency compared with reasonable controls. Cell viability is quantified and GFP turn-on is confirmed.

Minor comments below.

Figure 2d could be improved by a more clear SEM image, or by including a TEM image of the particles.

Overall transfection rates are fairly modest (doubling). Is this due to only using small quantities of particles per cell culture? Were larger particle densities tested? Perhaps some discussion could be added to teach readers how larger transfection rates as compared with controls could be accomplished?

Is there a reason A375-M cells were used for this work?

Reviewer 3 Report

The paper is very well written, the applied methods and the conclusions are quite appropriate. I would support further work along the lines presented.

Author Response

Thank you very much we keep on working on this system and hope for more progresses considering different DNA-vectors.